# Calibration of Sensor Network for Outdoor Measurement of PM2.5 on High Wood-Heating Smoke in Temuco City

Carlos Muñoz [1,*], Juan Huircan [1], Francisco Jaramillo [2] and Álex Boso [3]

1   Department of Electrical Engineering, Faculty of Engineering and Sciences, Universidad de La Frontera, Av. Francisco Salazar, Temuco 01145, Chile; juan.huircan@ufrontera.cl
2   Department of Electrical Engineering, Faculty of Physical and Mathematical Sciences, University of Chile, Av. Tupper, Santiago 2007, Chile; francisco.jaramillo@ing.uchile.cl
3   Department of Environment, CIEMAT, Avenida Complutense 40, 28040 Madrid, Spain; alex.boso@ciemat.es
*   Correspondence: carlos.munoz@ufrontera.cl

**Abstract:** In order to ascertain the spatial and temporal changes in the air quality in Temuco City, Chile, we created and installed a network of inexpensive sensors to detect PM2.5 particulate matter. The 21 measurement points deployed were based on a low-cost Sensiron SPS30 sensor, complemented with temperature and humidity sensors, an Esp32 microcontroller card with LoRa and WiFi wireless communication interface, and a solar charging unit. The units were calibrated using an airtight combustion chamber with a Grimm 11-E as a reference unit. The calibration procedure fits the parameters of a calibration model to map the raw low-cost particle-material measurements into reliable calibrated values. The measurements showed that the concentrations of fine particulate material recorded in Temuco present a high temporal and spatial variability. In critical contamination episodes, pollution reaches values as high as 354 μg/m$^3$, and at the same time, it reaches 50 μg/m$^3$ in other parts of the city. The contamination episodes show a similar trend around the city, and the peaks are in the time interval from 07:00 PM to 1:00 AM. In the winter, this time of day coincides with when families are usually home and there are low temperatures outside.

**Keywords:** air pollution; low-cost sensor; particle-matter calibration; spatial–temporal distribution

## 1. Introduction

In southern Chile, burning wood is widely valued as a primary energy source in the domestic sphere due to its low price compared to other heating alternatives [1]. However, its massive use and inefficient combustion generate high emissions of air pollutants. The city of Temuco, in recent years, has increased its levels of environmental pollution, with the highest levels of PM2.5 and PM10 [2–4] exceeding in both cases 300 μg/m$^3$. These reports show that every winter, the city of Temuco experiences numerous critical episodes due to poor air quality, with particulate matter that exceeds the national regulations and international standards, impacting public health and the quality of life of its residents. The leading cause of pollution by PM2.5 and PM10 in this city is the widespread use of wood-fired heaters due to their low cost and high availability in the area [1].

The city of Temuco is medium-sized in terms of population (approximately 304,000 inhabitants), with an urban area of 50 km$^2$ and a riverbed topography with surrounding hills. This city has two FEM (Federal Equivalent Methods) stations installed and operated by the Ministry of the Environment [5]. These two FEM stations are located in residential sectors separated by 4.5 km from each other. The air-quality measurements obtained through these FEM stations allow the adoption of measures to restrict the use of wood heating [3].

Becnel et al. [6] and Johnston et al. [7] state that FEMs can deliver reliable and accurate data to support decision-making; however, they are expensive to maintain. Due to the high costs, the number of stations is usually sparse, lacking the spatial resolution necessary to assess community exposure to PM2.5 particulate matter.

According to Johnston et al. [7], the impact of air pollution levels on health is dependent on pollutant concentrations and exposure levels. These factors vary at fine spatio-temporal scales in urban environments, driving the need for more data by increasing the number of sensor deployments and improving the sampling frequency. Alfano et al. [8] present an extensive review of the low-cost particulate matter sensors (LCPMSs) currently available on the market, their electronic characteristics, and their applications in the published literature and from specific tests. These LCPMSs are proposed as a complement to the measurements provided by the FEMs [9].

Among the LCPMSs described in the literature, there is a sensor node developed by Johnston et al. [10] that uses four low-cost devices (Alphasense OPC-N2, Plantower PMS5003 and PMS7003, Honeywell HPMA115S0). Johnston et al. [10] compared these LCPMSs to certified government stations, resulting in an RMS error of 6.065 and a Pearson coefficient of 0.878. Sayahi et al. [11] carried out a field evaluation of Plantower PMS1003 and PMS5003 sensors with reference equipment under different conditions, showing a good correlation in the winter season ($R^2 > 0.858$). Sayahi et al. [11] concluded that having various calibration factors for the same sensor model requires a systematic laboratory tuning or an on-site calibration strategy. Badura et al. [12] evaluated a network of 20 PMS5003 sensors, and the average values of the coefficient of determination ($R^2$) calculated between the sensor nodes and the government FEM stations were around 0.89 (with a range of 0.82 to 0.91, depending on the sensing device and station). Becnel et al. [6] proposed a platform with PMS3003 sensors made up of 50 nodes, which they deployed in an area of 100 km$^2$, obtaining a correlation of $R^2 = 0.88$ between the sensor nodes and the FEM. Tagle et al. [13] built a small network based on SDS011 particulate-matter sensors, which were compared with reference FEM stations, obtaining a correlation coefficient ($R^2$) between 0.47 and 0.86.

Ferrer-Cid et al. [14] state that PM2.5 particulate matter sensors, among others, are generally not calibrated by the manufacturer. If they have been, they lack calibration under the environmental conditions in which they will provide measurement data. To calibrate LCPMSs in the laboratory, Papapostolou et al. [15] and Sayahi et al. [16] developed calibration chambers that allow stable and reproducible gas and aerosol concentrations at low, medium, and high levels. Malings et al. [17] then uses an instrument laboratory standard, which has a combustion chamber to generate characteristic curves that allow adjustment of a specific calibration polynomial for each LCPMS. Magi et al. [18] proposed improving the correction with compensation of the measurements for humidity and temperature variations. Zaidan et al. [19] developed calibration systems to enhance the accuracy of LCPMSs, incorporating calibration models based on machine learning and virtual sensors. The methodology requires the construction of a database using continuous measurements of low-cost sensors and reference instruments. Then, it is analyzed to obtain information on atmospheric characteristics and air pollutants, which allows an understanding of the performance of the sensors in terms of consistency (relative to other sensors) and accuracy by comparison with the reference instruments. According to Ferrer-Cid et al. [14], incorporating low-cost sensors is a viable alternative to complement the measurements made at the FEM stations.

According to Datta et al. [20], using multiple LCPMSs allows obtaining information on intra-urban space–time variation, which allows extraction of sectorized information on PM2.5 pollution. In this sense, Schneider et al. [21] and Liu et al. [22] accounted for the non-uniformity of pollution through updated high-resolution maps of particulate pollution in the environment using 24 LCPMSs in the city of Oslo, Norway. A space-time analysis of PM2.5 particulate matter using multiple sensors in Dezhou City, China, Cao et al. [23] identified four stages of daily PM2.5 variation: accumulation, continuous pollution, dispersion, and cleaning, which is consistent with the pollution episodes generated by the use of heating.

Using mobile units makes achieving a flexible system to acquire information in a broad spatial spectrum possible. Still, comparing measurements taken in different parts of the city makes it difficult, since these are captured at deferred times. On the other hand, using measurements through LCPMSs is one of many mechanisms to achieve higher-

resolution pollution maps. Blanco et al. [24] and Quinteros et al. [25] carried out spatial characterization of particulate matter during winter nights in Temuco City using vehicles with GPS tracking that carried PM2.5 recording units. The investigation indicated that pollutants are distributed unevenly in Temuco City, revealing that in some neighborhoods, PM2.5 concentrations are almost twice as high as those measured at the government station.

The objective of the present study is to deploy and calibrate a low-cost sensor network platform in the city of Temuco, Chile. This sensor network allows for an analysis of the density of contaminants in the phases of accumulation, continuous contamination, dispersion, and cleaning during contamination episodes. The novelty of this paper and its main contribution is that we deployed a calibrated LCPM sensor network over a city with high-contamination episodes, allowing us to analyze the behavior of the pollutants over the entire high-contamination episode. This network will allow us to measure the pollutants in the city, evidencing the differences in particle-matter concentration within high-contamination episodes simultaneously in different sectors.

The paper is structured as follows: First, Section 1 furnishes an overview of the current research status on obtaining information on intra-urban space–time variation, which allows extraction of sectorized information on PM2.5 pollution, and depicts the objective of deploying and calibrating a low-cost sensor network platform in the city of Temuco, Chile. Then, in Section 2, we show the methods, and Section 3 focuses on depicting a novel calibration method for low-cost particle-matter sensors based on a nonlinear calibration model. Subsequently, in Section 4, we address the case study, presenting the deployment of the LCPM sensor network in Temuco City. As the main result, we show the spatio-temporal analysis of PM2.5 on three consecutive days, in which we can observe the accumulation of pollution and the washout due to rain. Lastly, in Section 5, the results and conclusions show the experience deploying this sensor network and calibration method, including limitations and future research directions.

## 2. Methods

Figure 1 depicts the method used, which involves two stages: calibration of the sensors in the laboratory with a reference instrument and deployment of the sensor nodes in the city. The following sections will describe each of the stages.

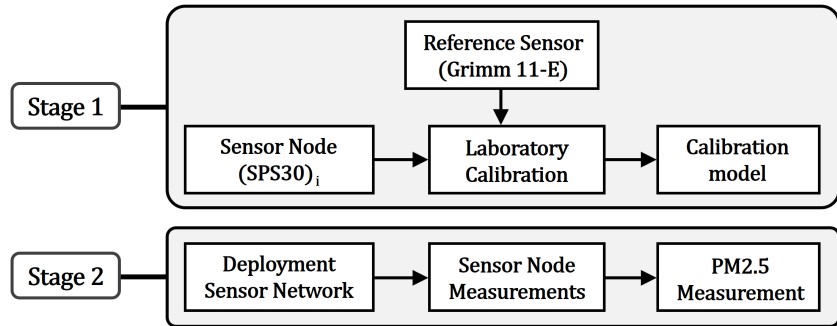

**Figure 1.** Methodology used in this work. In stage 1 the calibration setup was made and the calibration models were fitted. In stage 2, the sensor network was deployed and operated.

In the first stage, as the first step, the method uses a combustion chamber for plant material to compare the measurements of sensor nodes based on the Sensirion SPS30 device with a GRIMM 11-E reference instrument. Then, we apply calibration models to match the responses of the Sensiron SPS30 and those obtained with the GRIMM 11-E.

### 2.1. Particulate Matter IoT Devices

Each IoT device for measuring particulate matter has the structure indicated in Figure 2 and consists of the following components: LCPMS, temperature/humidity sensor, WiFi communication module, LoRa communication module, power for a 3.6 V Lithium battery,

7 W solar panel, and charge controller, all managed by an ESP32 microcontroller card. It has a storage unit type micro SD. All this is installed in a case with the IP67 standard.

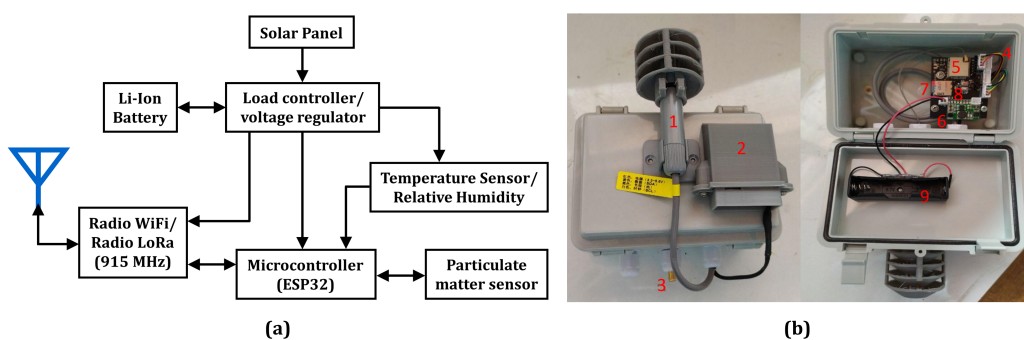

**Figure 2.** Sensor node of particulate matter. (**a**) Diagram. (**b**) Device, 1: Temperature Sensor / Relative Humidity, 2: Particulate Matter Sensor, 3: To the solar panel, 4: To the battery, 5: Microcontroller, Radio WiFi/Radio LoRa, 6: Load controller /Voltage regulator, 7: MicroSD Card , 8: Circuit board 9: Battery Holder for Li-Ion Battery

As the LCPMS, we used the Sensiron SPS30, a device for online measurements of particulate material PM1, PM2.5, PM4, and PM10 based on light scattering. This device presents an excellent performance Bulot et al. [26] and has a more compact package than the alternatives. The net measurement is converted into units of mass concentration (0–1000 µg/m$^3$) [27]. The device supports operating temperatures between −10 and 60 °C and humidity up to 95% RH, which is compatible with the range of temperatures measured in the City of Temuco (−5 to 38 °C) [28], and during periods of cold weather and without precipitation, the humidity reaches 95%. When precipitation occurs, the device turns off because there are no restrictions on using wood-fired heaters since the particulate material settles with the rain. According to the manufacturer, PM2.5 has a mass concentration accuracy of ±10 µg/m$^3$ for a range of 0 to 100 µg/m$^3$. On the other hand, for ranges between 100 and 1000 µg/m$^3$, the accuracy is ±10% of the measured value. Furthermore, according to the manufacturer, the SPS030 has an estimated lifespan of 10 years and includes a self-cleaning mechanism.

Additionally, we programmed the sensor node with a data-acquisition application, including programmable sampling time, local storage in the micro SD memory, tools to update the software remotely, and an MQTT protocol to upload the collected data online to a mySQL database. We used the ThingsBoard open-source IoT platform for real-time data monitoring.

### 2.2. Reference Instrument

We used the GRIMM 11-E equipment to acquire the reference material particulate measurements. This instrument complies with the EN12341 [29] and EN14907 [30] standards for PM10 and PM2.5 measurement, respectively. It has a counting range of 1–2,000,000 particles/L and has a measurement range from 0.1 µg/m$^3$ to 6 mg/m$^3$ with reproducibility of ±3% over the entire measurement range.

### 3. Sensor-Node Calibration

With the purpose of calibrating the LCPMSs, we designed and implemented a controlled laboratory experiment. In this experiment, an airtight chamber with forced airflow was prepared to generate the combustion of a specific vegetable mass inside. Two LCPMSs and a reference instrument (GRIMM 11E) were installed to register PM2.5 measurements, as shown in Figure 3a.

The three sensors were configured to be synchronized with the start of the PM2.5 measurements. The sampling time for data acquisition was set at 6 s, where the recorded data were stored in the internal memory of the LCPMS and the reference instrument. In

addition, the devices were always on during the experiment. Finally, the total length of the experiment was set at 8 h.

The data registered by the three sensors were preprocessed using a technique derived from Coleman and Meggers [31]. In our case, we considered a time window of 30 s (i.e., five measurements), where the median value for each time window was extracted, thus reducing the sampling time (to 30 s) and the quantity of measured data (930 data points). Figure 3b shows the variation in time of the PM2.5 concentration after preprocessing. The measured concentration exceeds the specified upper limit for each instrument to finally reach a minimum value at the end of the experiment. The yellow curve in Figure 3b represents the PM2.5 measurements acquired by the reference instrument used to calibrate both LCPMSs considered in this work (LCPMS 1, blue curve; LCPMS 2, red curve).

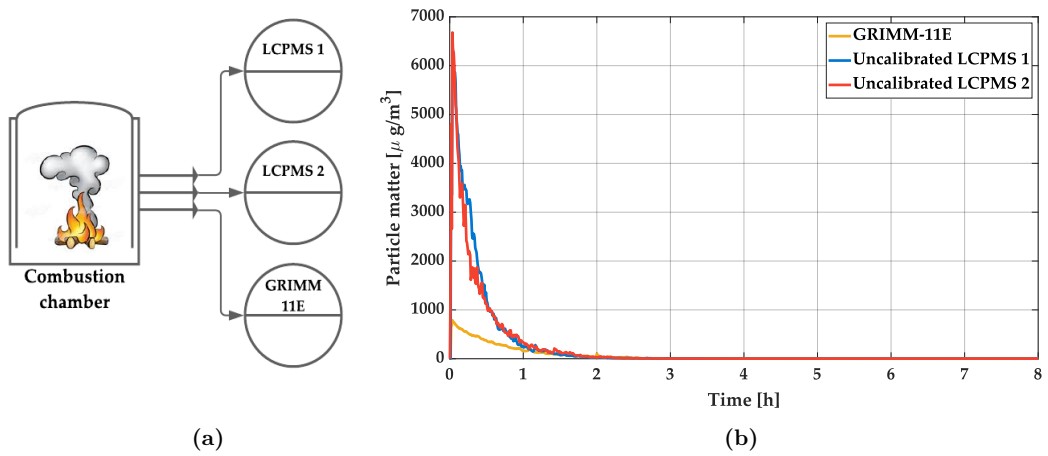

(**a**) (**b**)

**Figure 3.** (**a**) Calibration experiment scheme. (**b**) Acquired signals from the reference instrument (yellow line) and the LCPMSs (blue and red lines).

The next step in the calibration procedure involves finding a calibration model $y_c(x, \theta)$ capable of mapping the raw LCPMS measurements onto reliable calibrated values. In this work, we defined a cost function that should be minimized using the data extracted from the laboratory experiment and a proposed calibration model to find the optimum set of parameters of $y_c(x, \theta)$. The cost function is detailed in Equation (1), where $y_c$ is the calibration model and $x_1$, $x_2$, and $y$ are the data related to LCPMS 1, LCPMS 2, and GRIMM 11E, respectively.

$$\theta = \arg\min_{\theta} \left( \sum_{i=1}^{N} [y_c(x_1(i), \theta) - y(i)]^2 + \sum_{i=1}^{N} [y_c(x_2(i), \theta) - y(i)]^2 \right) \tag{1}$$

### 3.1. Polynomial Calibration Model

In references [32,33], polynomial functions are one of the commonly proposed calibration models; due to this, we consider as $y_c(x, \theta)$ a polynomial model of order $p$, indicated in Equation (2).

$$y_c(x, \theta) = \sum_{i=1}^{p} \theta_i \cdot x^{i-1} \tag{2}$$

By using the MATLAB optimization toolbox, the cost function in Equation (1) was minimized considering different values for $p$ to the proposed model in Equation (2). It is important to mention that, before the optimization, a second preprocessing step was applied to the experimental data for outlier removal. As a result, the optimum order of the polynomial was $p = 9$, and the optimum parameters are detailed in Table 1. Moreover, Figure 4 shows three graphs that complement the results achieved by the proposed calibration process. At first, Figure 4a displays the optimum calibration model for LCPMS

1/LCPMS 2. Then, Figure 4b presents the error histogram between LCPMS 1/LCPMS 2 and the calibrated model, where the standard deviation of the calibration error is equal to 7.5904. Finally, Figure 4c compares the experimental data registered from the reference instrument (GRIMM-11E) and the calibrated LCPMS 1/LCPMS 2.

**Table 1.** Optimum parameter set $\theta$ with 95% confidence bounds for the polynomial calibration model.

| Coefficient ($\theta$) | Value | 95% Confidence Bounds |
|:---:|:---:|:---:|
| $\theta_1$ | 111.9 | (110.6, 113.1) |
| $\theta_2$ | 411.6 | (408.1, 415) |
| $\theta_3$ | $-487$ | $(-504.2, -469.9)$ |
| $\theta_4$ | 418.8 | (392.3, 445.3) |
| $\theta_5$ | $-199.8$ | $(-217.3, -182.4)$ |
| $\theta_6$ | 54.99 | (49.03, 60.95) |
| $\theta_7$ | $-9$ | $(-10.15, -7.854)$ |
| $\theta_8$ | 0.8655 | (0.7405, 0.9905) |
| $\theta_9$ | $-0.04517$ | $(-0.0524, -0.03794)$ |
| $\theta_{10}$ | 0.000987 | (0.0008149, 0.001159) |

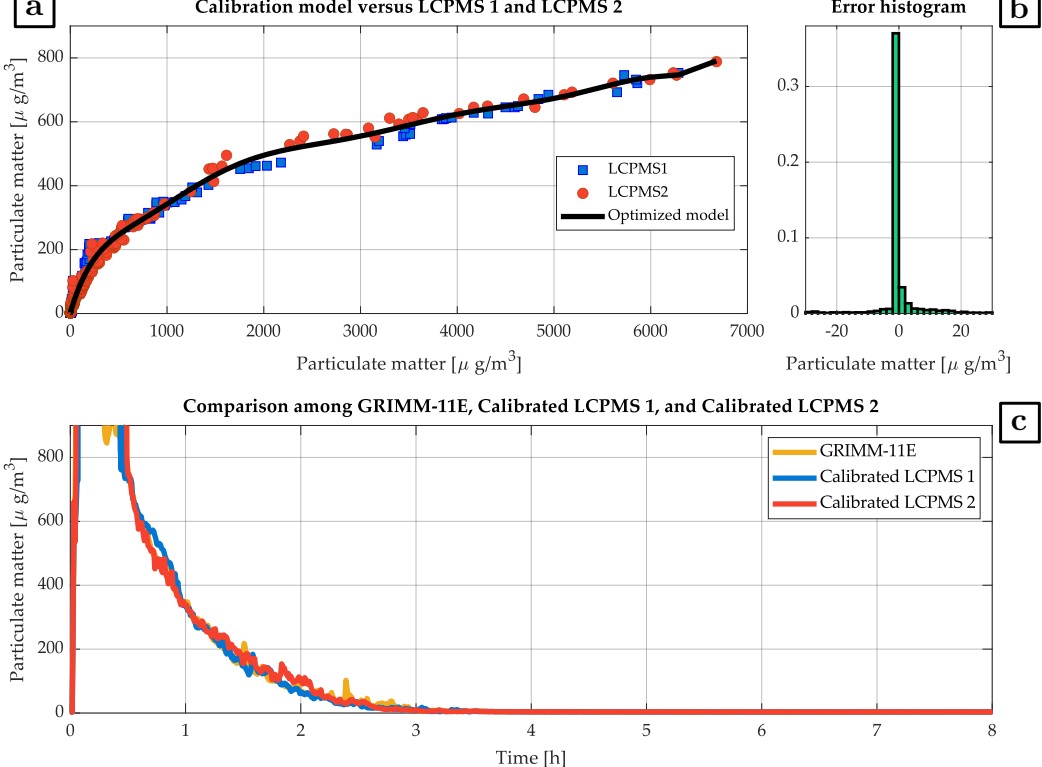

**Figure 4.** (**a**) Ninth-order polynomial calibration model for LCPMS 1 and LCPMS 2. (**b**) Calibration error histogram. (**c**) Comparison among GRIMM-11E and the calibrated LCPMS.

### 3.2. Nonlinear Function Calibration Model

To avoid overfitted and over-parameterized calibration models, we propose a nonlinear function composed of two exponential components and one square root term. This nonlinear model is detailed in Equation (3):

$$y_c(x, \theta) = a_0 \cdot e^{b_0 \cdot x} + a_1 \cdot e^{b_1 \cdot x} + c_0 \cdot \sqrt{x} \; , \tag{3}$$

where the parameter vector $\theta$ is represented by $\theta = \begin{bmatrix} a_0 & b_0 & a_1 & b_1 & c_0 \end{bmatrix}$.

After applying the second preprocessing step for outlier removal and using the MAT-LAB optimization toolbox, the cost function in Equation (1) was minimized considering the nonlinear model (Equation (3)). The resultant optimum set of parameters $\theta$ is detailed in Table 2.

**Table 2.** Optimum parameter set $\theta$ with 95% confidence bounds for the nonlinear calibration model.

| Coefficient ($\theta$) | Value | 95% Confidence Bounds |
|:---:|:---:|:---:|
| $a_0$ | 193.8 | (192.7, 194.8) |
| $b_0$ | $5.989 \times 10^{-5}$ | $(5.901 \times 10^{-5}, 6.076 \times 10^{-5})$ |
| $a_1$ | $-198.8$ | $(-199.8, -197.7)$ |
| $b_1$ | $-0.001553$ | $(-0.001559, -0.001546)$ |
| $c_0$ | 5.969 | (5.939, 5.999) |

In the same manner, as shown above, Figure 5 helps to understand better the scope of the results achieved by the proposed calibration process. Figure 5a displays the optimum calibration model for the LCPMS 1/LCPMS 2. Then, Figure 5b presents the error histogram between LCPMS 1/LCPMS 2 and the calibrated nonlinear model, where the standard deviation of the calibration error equals 6.4562. Finally, Figure 5c compares the experimental data registered from the reference instrument (GRIMM-11E) and the calibrated LCPMS 1/LCPMS 2.

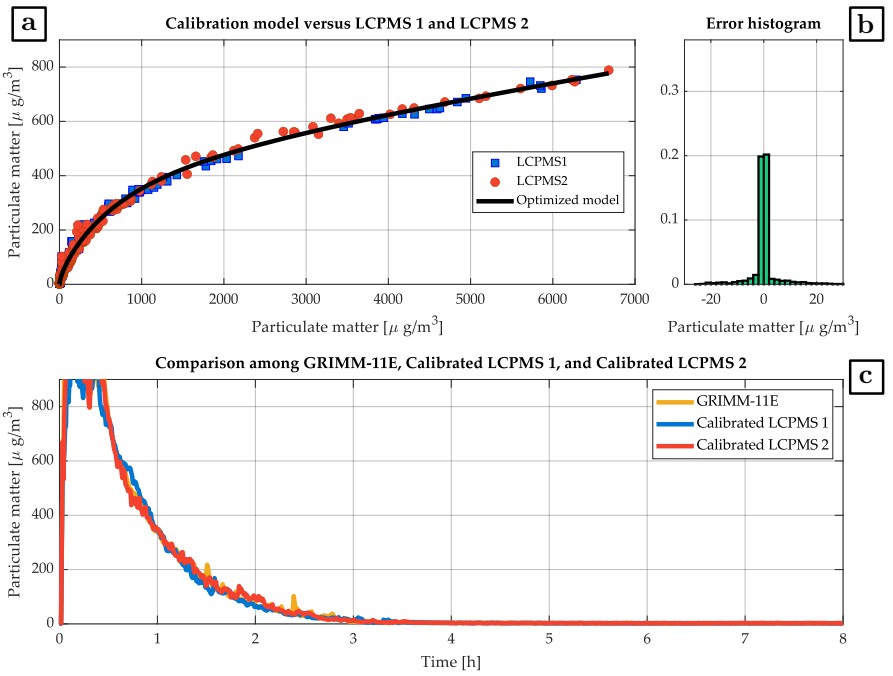

**Figure 5.** (**a**) Nonlinear calibration model for LCPMS 1 and LCPMS 2. (**b**) Calibration error histogram. (**c**) Comparison among GRIMM-11E and the calibrated LCPMSs.

Given the calibration results (Figures 4 and 5), both proposed models achieved an adequate mapping from the raw data to the calibrated data. However, the nonlinear calibration model was the one that obtained the best results as to the standard deviation of the calibration error and lower number of parameters. This last is crucial to obtain generalized models instead of overfitted models.

## 4. Case Study: IoT Network for Particle-Size Monitoring in Temuco City

### 4.1. Comparison of the LCPM with Other Solutions

For a comparative analysis of the LCPM device with respect to other sensor devices, we analyzed characteristics such as the type of use (indoor, outdoor), price, range, precision, and communication. The outdoor property is relevant because the sensor network is intended to acquire the PM10 particle-matter behavior outdoors. The communication is a requirement to real-time collect the data acquired in a central database. The cost per unit is relevant to deploy a significant number of units to give an insight into the pollution behavior in the city through time. Table 3 compares these sensors' characteristics, and as shown, the LCPM fulfills the requirements issued for the sensor network deployment. In the case of GRIMM 11-E, the precision was not available in the datasheet, but the vendor stated a reproducibility of $\pm 3\%$ over the total measuring range. We calibrated the LCPM devices in the laboratory, coded and installed the data-collection system with the respective dashboards, and finally proceeded with the next step: deploying the sensor network in the city.

**Table 3.** Comparison of particle-matter sensors.

| | Sensor Type | | |
|---|---|---|---|
| Features | Low-Cost Node Sensors Proposed | Low-Cost Node Sensors Commercially Available | GRIMM 11-E or Similar |
| Cost per unit USD$ | 120 | 200–500 [34,35] | 10,000–20,000 [35] |
| PM2.5 | yes | yes | yes |
| PM10 | yes | yes | yes |
| Range | 0–1000 µg/m$^3$ | 0–1000 µg/m$^3$ | 0.1–6000 µg/m$^3$ |
| Precision | <100 µg/m$^3$: $\pm$ 10 µg/m$^3$ | <100 µg/m$^3$: $\pm$ 10 µg/m$^3$ | N.A. |
| Particle Range Size | 0.3 to 2.5 µm | N.A. | 0.25–31µm |
| Wireless Connectivity | LoRa, WiFi | Global cellular 2G/3G/4G | Bluetooth |
| Use Type | Outdoor | Outdoor | Laboratory |
| Additional Measurement | Temperature, Humidity | Temperature, Humidity | N.A. |

### 4.2. Node-Deployment Criteria

We deployed a network of 21 IoT devices with calibrated LCPMSs for measuring particulate matter in Temuco City, as shown in Figure 6.

For the location of the nodes, the following aspects were considered: that the place is far from a source of smoke emission, that there is access to the installation site, the availability of connectivity, the availability of energy, and the uniform distribution of the equipment. This last element is not necessarily actual since the x-distant point to an already installed node was not necessarily available to install the next node. In the case of residential nodes, we considered the participation of citizens, who agreed to provide space for the installation in the backyard of their homes and connectivity for the required nodes. In the case of the nodes installed in public distributions, the authority provided access to data and energy connectivity. We installed the nodes at outdoor sites more than 30 m from smoke emission sources (stove or boiler chimneys) and around 2 m in height. The selected 2 m is because the average size of people is between 1.5 m and 1.8 m. In addition, this height allows us to install the solar panel on a nearby roof.

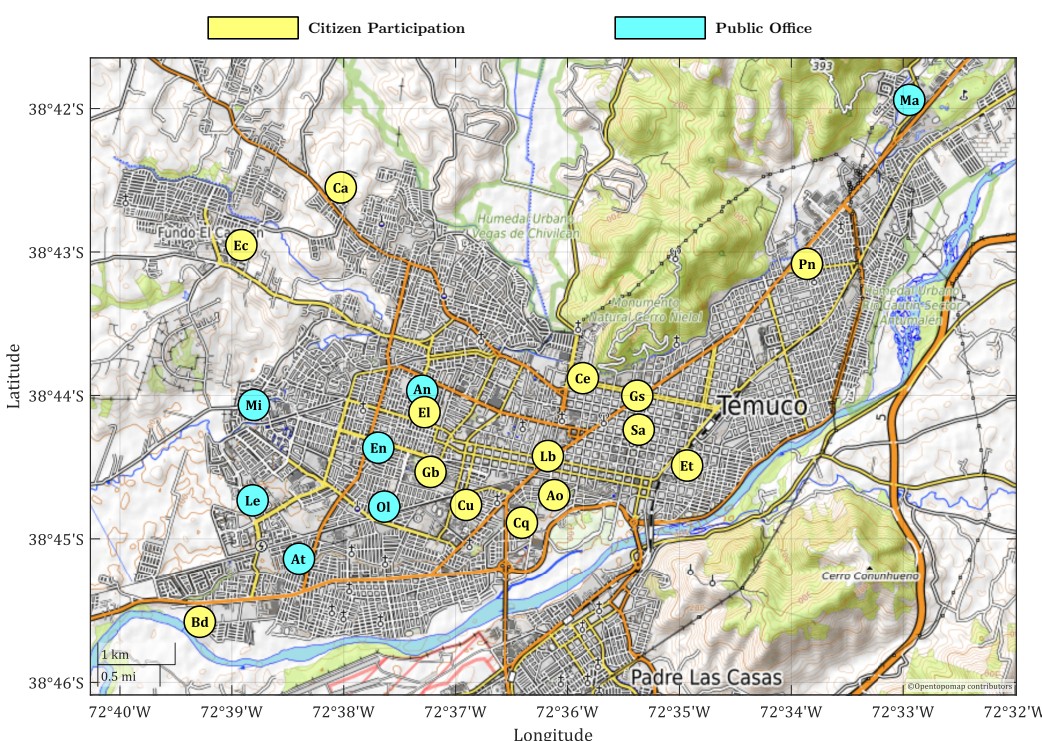

**Figure 6.** Temuco map, 14 July 2020 at 21:00 h (adapted from OpenStreetMap [36]).

**Table 4.** Node information.

| Node Name | Abbreviation | Type |
|---|---|---|
| Altamira | At | Citizen Participation |
| Altos de Mirasur | Mi | Citizen Participation |
| Andes | An | Citizen Participation |
| Aseo y Ornato | Ao | Public Office |
| Bodega Droguería | Bd | Public Office |
| CECOSF Arquenco | Ca | Public Office |
| CECOSF Las Quilas | Cq | Public Office |
| Cementerio | Ce | Public Office |
| Ciencias Físicas UFRO | Cu | Public Office |
| El Carmen | Ec | Public Office |
| El Trencito | Et | Public Office |
| Entrelagos | En | Citizen Participation |
| Escuela Llaima | El | Public Office |
| Galo Sepúlveda | Gs | Public Office |
| German Becker | Gb | Public Office |
| La Lechería | Le | Citizen Participation |
| Las Mariposas | Ma | Citizen Participation |
| Liceo Bicentenario | Lb | Public Office |
| Olimpia | Ol | Citizen Participation |
| Pueblo Nuevo | Pn | Public Office |
| Smart Araucanía | Sa | Public Office |

The calibrated LCPMSs were configured to capture information every 10 min. To compare the PM25 measurements in different sectors every hour, we computed the average hourly values in every station. To identify the areas in which the nodes have been installed, identifiers similar to those proposed in [37–39] were used. Since there is no single urban zoning identification, we associated the deployment of the nodes with two different types of locations, citizen participation and public office. Citizen participation refers to citizens who put the location to install sensor nodes at our disposition. In this case, the nodes were deployed in the backyards of residential homes powered by internal batteries and a small solar panel. These residential nodes are connected to the home's WiFi to deliver the sensed data. Public office refers to public schools, family health centers, and municipal warehouses. The nodes were plugged into the local electricity network and connected to the data centers via Ethernet wires. Figure 6 shows the deployment of the nodes by type in the city of Temuco, and Table 4 shows the characteristics of the nodes installed.

### 4.3. Results of Three Consecutive Days

The concentrations of fine particulate material recorded in Temuco show an evident temporal variability, with the maximum attention in the time slot from 7:00 p.m. to 1:00 a.m. This time of day coincides with when families are usually at home and temperatures outside are low. Regarding home emissions associated with firewood, various studies have shown a high dependence between the temperature, a high energy demand for heating in homes with poor thermal insulation, and the emission rate of pollutants [3]. This relationship is observed in the differences in levels of particulate matter found in our sensors. Some residential neighborhoods with the highest population density and poorest housing quality show a higher emission of air pollutants associated with firewood, especially during peak hours. On the other hand, the sensors indicate that the highest concentrations, with emergency levels for PM2.5, occur in areas around the geographic center of the city, as can be seen in Figures 6 and 7. From the center and moving towards the periphery, the average levels of particulate matter decrease.

This situation can be due to the geographical distribution of the city, which is in a valley encased by two hills and a river. When the wind blows, it tends to displace pollutants in the direction of the river.

Figures 7–9 show the results of monitoring on three consecutive days (14 June, 15 June, and 16 June). These figures show that days 14 and 15 were cold, with a minimum temperature below 5 °C and low humidity. However, on the 17th, it rained, the temperature rose, and pollution diminished throughout the city. We want to note that the 15th day was the most critical since the contamination had accumulated for two consecutive days, reaching critical thresholds in many stations at peak times (21:00–03:00). In some stations, these peaks were more extreme, reaching values as high as 350 μg/m$^3$, which greatly exceeded all permitted pollution standards for many consecutive hours. This situation could be explained due to many wood stoves heating homes when temperatures are freezing and people are home. However, given the geographical characteristics of the city, where a natural wind corridor along the river exists, there is an excellent cleaning capacity. The figures show that at some off-peak hours, the city shows medium to low pollution rates in all the stations. From Figures 7–9 we see that only some stations show extreme PM10 values, and the places that reach the highest pollution values are near the city's center, which is geographically located where the wind corridor enclosed by two hills narrows (see Figure 6). An analysis with more data is required to determine how the topography in different climatic conditions affects the distribution of particulate-matter pollution of PM2.5 and PM10 over the city.

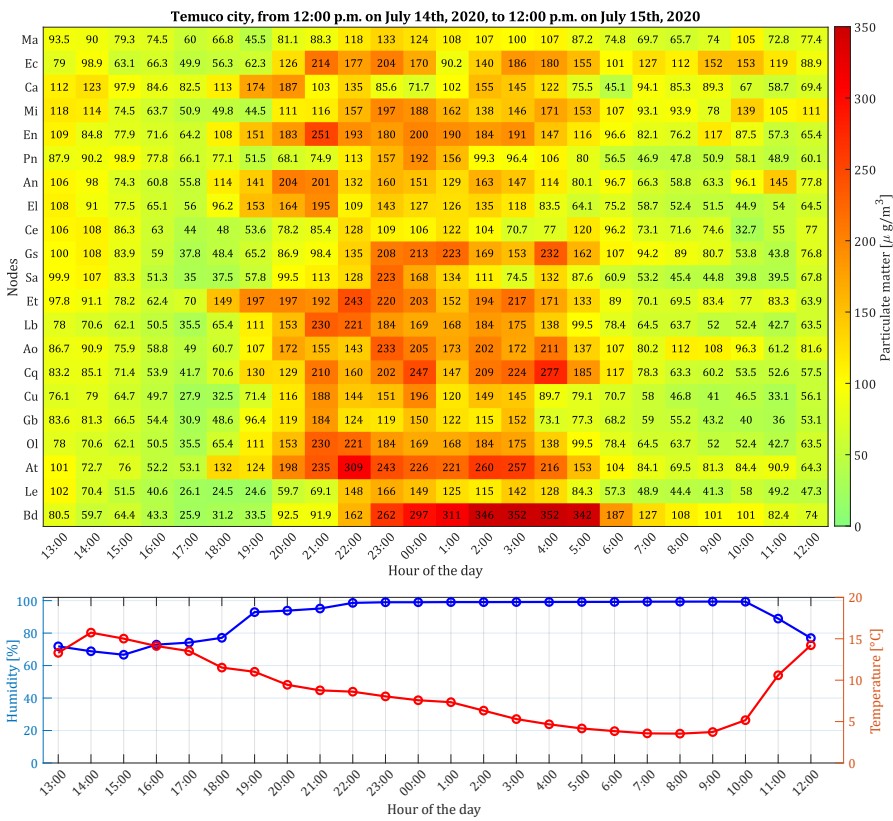

**Figure 7.** Nodes and particulate matter on 14 July 2020. Red line: Temperature, Blue line: Relative humidity.

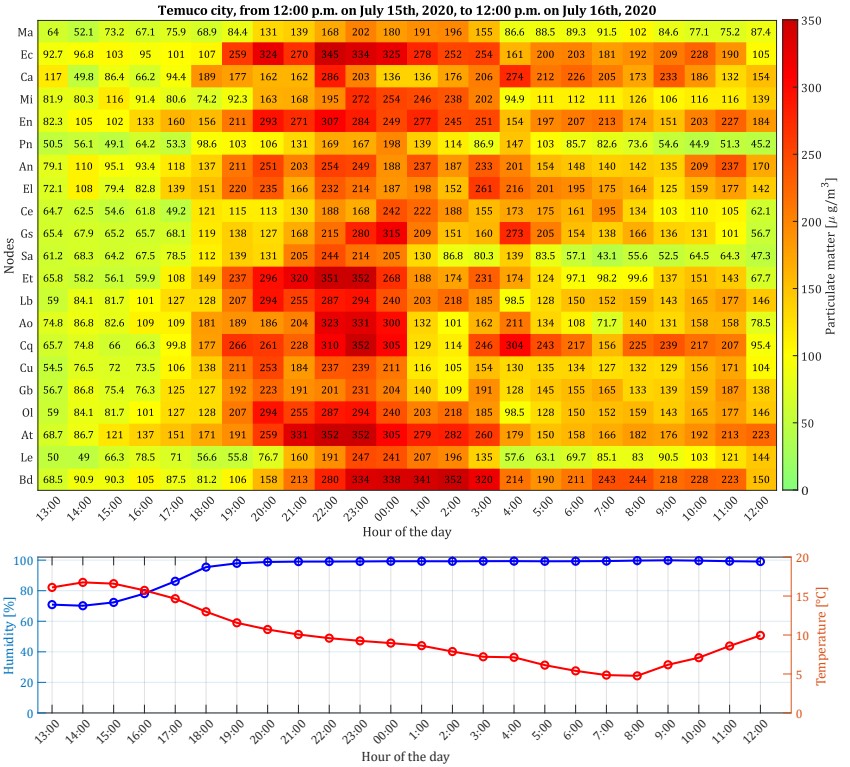

**Figure 8.** Nodes and particulate matter on 15 July 2020. Red line: Temperature, Blue line: Relative humidity.

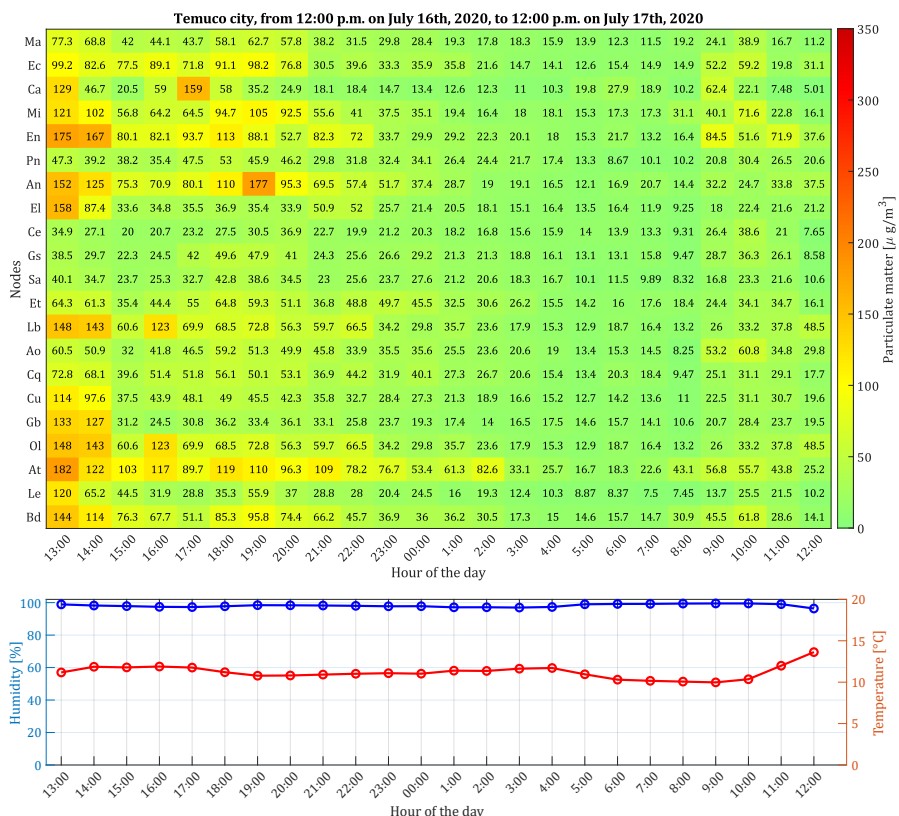

**Figure 9.** Nodes and particulate matter on 16 July 2020. Red line: Temperature, Blue line: Relative humidity.

## 5. Discussion and Conclusions

The objective of this study was to deploy and calibrate a low-cost particulate-matter sensor (LCPMS) network platform in Temuco City, Chile. This sensor network allows for analysis of the density of contaminants in the phases of accumulation, continuous contamination, dispersion, and cleaning during contamination episodes. The main contribution of this paper is to review the dynamic behavior of the pollutants in the whole city over the entire event of a high-contamination episode. This spatio-temporal analysis provides an insight into how the contaminants move in Temuco City over time during episodes of high contamination and cleansing due to rain. In Latin America and, in general, in low- and middle-income countries, low-cost sensors can revolutionize traditional monitoring systems since they provide air-quality information in real time, with excellent spatial precision and, when calibrated, with a remarkable level of validity and reliability. Their low-cost nature, especially when compared to official monitoring stations, gives this technology the potential to democratize air pollution information, making it accessible to a broad audience. The development of low-cost sensor-based monitoring systems constitutes an opportunity to give voice to, amplify, and represent local needs, especially those of socially vulnerable groups. To achieve this, it will be necessary that, in addition to technological progress such as those shown in this study, researchers and environmental authorities coordinate their work with volunteers, representatives of non-governmental organizations, and community groups.

Properly selecting the calibration methods for low-cost pollution sensors is relevant to acquire reliable data worth making a spatio-temporal analysis during contamination episodes.

Regarding the calibration methods shown in this work, we proposed a nonlinear calibration model, which was compared with a polynomial calibration model seen in the state of the art [40–43]. The proposed nonlinear model obtained a lower standard deviation of the calibration error (6.4562) than the polynomial model (7.5904). Furthermore, the nonlinear model includes five parameters instead of the nine parameters of the polynomial model. This smaller number of parameters may make the model easier to calibrate and less susceptible to overfitting, which is consistent with the goals stated

by Warder and Piggott [44]. The LCPM sensors deployed have been working outdoors, enduring rough operating conditions such as rainy days and temperatures lower than minus 5 °C. The calibration can be affected by the intrusion of little spiders and dust to form tartar stuck inside the measuring chamber. We advise recalibrating the LCPM sensors at least once a year. The amount of LCPM sensors deployed depends on topography factors, population density, and the network coverage to deploy a sensor network that provides enough spatio-temporal information to understand the characteristics of the contamination episodes in the target location. This study showed that relevant data to understand the movement of spatio-temporal behavior of the contaminants could be acquired using calibrated LCPM sensor networks. The future challenge is to keep running this LCPM sensor network, increasing the number of nodes and replicating this setup in other cities with their features, such as population density, network coverage, and topography. It is relevant to study strategies to keep involving the city's inhabitants in the issue of deploying the sensor network with physical spaces for LCPM sensor installation and using their home's internet connectivity.

**Author Contributions:** Conceptualization, C.M., F.J., and J.H.; methodology, C.M., F.J., and J.H.; software, C.M., F.J., and J.H.; validation, C.M., F.J., Á.B, and J.H.; formal analysis, C.M., F.J., Á.B., and J.H.; investigation, C.M., F.J., and J.H.; resources, C.M., F.J., and J.H.; data curation, C.M., F.J., and J.H.; writing–original draft preparation, C.M., F.J., and J.H.; writing–review and editing, C.M., F.J., and J.H.; visualization, C.M., F.J. , Á.B., and J.H.; supervision, C.M., F.J., and J.H.; project administration, C.M., F.J., and J.H.; funding acquisition, C.M., F.J., and J.H. All authors have read and agreed to the published version of the manuscript.

**Funding:** This research was funded by grant number IDI18-0003, BID-FOMIN-LAB, SMARTCITY IN A BOX, the ANID/FONDECYT 1220178, and ANID—Basal funding for Scientific and Technological Center of Excellence, IMPACT (Center of Interventional Medicine for Precision and Advanced Cellular Therapy), FB210024.

**Acknowledgments:** The authors thank Rodrigo Fuentes Inzunza from Universidad de Concepción for allowing us to use the GRIMM 11-E equipment for acquiring the reference material particulate measurements used in the LCPM calibration procedures.

**Conflicts of Interest:** The authors declare no conflict of interest.

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
