# Peer review of "Calibration of Sensor Network for Outdoor Measurement of PM2.5 on High Wood-Heating Smoke in Temuco City"

_processes, doi:10.3390/pr11082338_

Round 1
Reviewer 1 Report
The authors have taken extensive efforts to design and install a low cost sensor network across Temuco City to assess the spatial distribution of PM2.5 pollutants. I would like to raise a few queries for the authors before publication
1) Line 1 of Abstract is not properly stated as it failes to mention sensor network. The wordings suggested as below
"In order to ascertain the spatial and temporal change of the air quality in Temuco City, Chile, we created and installed a network of inexpensive sensors to detect PM2.5 particulate matter".
2) Line 17- Its preferable if authors provide the values of PM2.5 and PM10.
3) Line 123=EN12341 and EN14907, provide the standards they refer to for the ease of readers.
4) In Abstract provide a brief average values of PM2.5 or the peak PM 2.5 values during the critical days of study.
5) Line 114, the accuracy range for installed sensor is mentioned for values upto 100µg/m3. What is the expected accuracy beyond the range.
6) Line 125 mentions the measurement range of GRIMM 11E as 0.1µg/m3 125 to 6µg/m3. If so then why graph 3a shows its reading beyond the range. Also the authors need to mention the measuring range for installed sensor.
Author Response
Response to Reviewer 1 comments
The authors have taken extensive efforts to design and install a low cost sensor network across Temuco City to assess the spatial distribution of PM2.5 pollutants. I would like to raise a few queries for the authors before publication.
Point 1: Line 1 of Abstract is not properly stated as it failes to mention sensor network. The wordings suggested as below
"In order to ascertain the spatial and temporal change of the air quality in Temuco City, Chile, we created and installed a network of inexpensive sensors to detect PM2.5 particulate matter".
Response 1: The manuscript has been changed to include the reviewer’s suggestion.
Point 2: Line 17- Its preferable if authors provide the values of PM2.5 and PM10.
Response 2: The following text has been added (line 20).: “exceeding in both cases 300 µg/m3.”
Point 3: Line 123=EN12341 and EN14907, provide the standards they refer to for the ease of readers.
Response 3: Both standards were added as bibliographical references.
Point 4: In Abstract provide a brief average values of PM2.5 or the peak PM 2.5 values during the critical days of study.
Response 4: We have changed a paragraph in the abstract to a more descriptive one:
“The measurements showed that the concentrations of fine particulate material recorded in Temuco present a high temporal and spatial variability. In critical contamination episodes, pollution reaches values as high as 354 $\mu g/m^{3}$, and at the same time, it reaches 50 $\mu g/m^{3}$ in other places of the city. The contamination episodes show a similar trend along the city, and the peaks are in the time interval from 07:00 PM to 1:00 AM.”
Point 5: Line 114, the accuracy range for installed sensor is mentioned for values up to 100µg/m3. What is the expected accuracy beyond the range.
Response 5: The following sentence has been included in blue color (line 131): “On the other hand, for ranges between 100 and 1000 μg/m3, the accuracy is ± 10% of the measured value.”
Point 6: Line 125 mentions the measurement range of GRIMM 11E as 0.1µg/m3 to 6µg/m3. If so then why graph 3a shows its reading beyond the range. Also the authors need to mention the measuring range for installed sensor.
Response 6: The manuscript had a spelling mistake in the range unit. The revised manuscript includes this correction (line 142) “0.1 µg/m3 to 6 mg/m3”, marked in blue.

Reviewer 2 Report
The article entitled "Calibration of sensor network for outdoor measurement of PM2.5 on high wood-heating smoke in Temuco City " presents the implementation of a low-cost PM2.5 particulate matter to determine the spatial and temporal variation of the air quality in Temuco City. In addition, some measurements were performed on both the low-cost sensor and an airtight combustion chamber with a Grimm 11-E for the calibrating process.
In my opinion, the paper title could be modified to also considered the implementation of the low-cost sensor and not only the calibration process. The abstract and conclusions also seem to be oriented to propose a new sensor.
The results are promising even if the following observations need to be made:
1) Insert a comparative new Table with the costs of the proposed sensors including also the non-low-cost ones (for example GRIMM 11-E).
2) Insert in the same Table the comparisons between the sensors, and their main characteristics to make readers understand which performances are not able to reach the low-cost sensors.
3) It is necessary to present the limitations of the proposed calibration method.
4) Limitations of this study should be discussed, especially regards the application in other contests.
Minor editing of the English language is required
Author Response
Response to Reviewer 2 comments
Comments and Suggestions for Authors
The article entitled "Calibration of sensor network for outdoor measurement of PM2.5 on high wood-heating smoke in Temuco City " presents the implementation of a low-cost PM2.5 particulate matter to determine the spatial and temporal variation of the air quality in Temuco City. In addition, some measurements were performed on both the low-cost sensor and an airtight combustion chamber with a Grimm 11-E for the calibrating process.
The results are promising even if the following observations need to be made:
Point 1: In my opinion, the paper title could be modified to also considered the implementation of the low-cost sensor and not only the calibration process. The abstract and conclusions also seem to be oriented to propose a new sensor.
Response 1: We consider that including the implementation of the low-cost sensor in the title corresponds to a “major revision” because if we change the title, we also should include a more extended description of the technical details of the deployment process of each node and the whole network sensor. On the other hand, the abstract and conclusions have been updated for a better understanding.
Point 2: Insert a comparative new Table with the costs of the proposed sensors including also the non-low-cost ones (for example GRIMM 11-E). Insert in the same Table the comparisons between the sensors, and their main characteristics to make readers understand which performances are not able to reach the low-cost sensors.
Response 2: A new table (Table 3) has been added in the subsection: Comparison of the LCPM with other solutions, with the following description:
“For a comparative analysis of the LCPM device with respect to other sensor devices, we analyzed characteristics such as the type of use (indoor, outdoor), price, range, precision, and communication. The outdoor property is relevant because the sensor network is intended to acquire the PM10 particle matter behavior in the outdoors. The communication is a requirement to real-time collect in a central database the data acquired. The cost per unit is relevant to deploy a significant number of units to give an insight into the pollution behavior in the city through time. Table \ref{tab:LCSComp} compares these sensors' characteristics, and as shown, the LCPM fulfills the requirements issued for the sensor network deployment. In the case of GRIMM 11-E, the precision was not available in the datasheet, but the vendor informed a reproducibility of $\pm 3\%$ over the total measuring range. We calibrated the LCPM devices in the laboratory, coded and installed the data collection system with the respective dashboards, and finally proceeded with the next step: deploying the sensor network in the city.”
|
|
Sensor Node Type |
||
|
Features |
Low-cost Node sensors proposed |
Low-cost Node sensors commercially available |
GRIMM 11-E or similar |
|
Cost per unit USD$ |
120 |
190 - 500 [3][4] |
10,000- 20,000 [4] |
|
PM2.5 |
yes |
yes |
yes |
|
PM10 |
yes |
yes |
yes |
|
Range |
0 -1000 µg/m3 |
0 - 1000 µg/m3 |
0.1 - 6000 µg/m3 |
|
Precision |
< 100 µg/m3: ± 10 µg/m3 |
< 100 µg/m3: ± 10 µg/m3 |
N.A. |
|
Particle Range size |
0.3 to 2.5 µm |
N. A. |
0.25 - 31µm |
|
Wireless Connectivity |
LoRa, WiFi |
Global cellular 2G/ 3G/ 4G |
Bluetooth |
|
Use Type |
Outdoor |
Outdoor |
Laboratory |
|
Additional Measurement |
Temperature, Humidity |
Temperature, Humidity |
N.A. |
Point 3: It is necessary to present the limitations of the proposed calibration method.
Response 3: This additional text has been included at the end of the conclusions section:
“The LCPM sensors deployed have been working outdoors, enduring rough operating conditions, such as rainy days and temperatures lower than minus 5 °C. The calibration can be affected by the intrusion of little spiders and dust stuck to form tartar inside the measuring chamber. We advise recalibrating the LCPM sensors at least once a year.”
Point 4: Limitations of this study should be discussed, especially regards the application in other contexts.
Response 4: This additional text has been included at the end of the conclusions section:
“The amount of LCPM sensors deployed depends on topography factors, population density, and the network coverage to deploy a sensor network that provides enough spatio-temporal information to understand the characteristics of the contamination episodes in the target location.”
Point 5: Comments on the Quality of English Language. Minor editing of the English language is required.
Response 5: The new version of our manuscript has been thoroughly reviewed to improve the writing.

Reviewer 3 Report
Dear Collegues,
The idea on which the paper is based is simple and not new as multi-parameter environmental control units are widespread in many cities. In fact, many of the political choices that affect citizens' lives depend on their measurements, such as the closure to traffic of cities or parts of the city with limited traffic areas or the adoption of alternating license plates, etc. Nevertheless, the technical challenge faced by this work from an electronic point of view with the development of a new economic control unit, from a more measuring point of view, with the management of the calibration of the system and from a less prosaic point of view of the collection and management of data coming from many control units distributed in the city for a long time, must certainly be rewarded. I add that the article is very clear, never boring, very well balanced in its subsections. The bibliography is valid and it is in particular for the measurement aspects related to the calibration of these control units which are by no means trivial. Even the technical choices relating to the sensor nodes fall within the state of the art of these devices.
As far as I'm concerned, the article can be accepted in this way, from a purely academic point of view, I allow myself to point out three publications that could be of interest to the Authors and to be reasonably considered for the bibliography in the introduction:
Pasquali, V., D'Alessandro, G., Gualtieri, R., & Leccese, F. (2017). A new data logger based on raspberry-pi for arctic notostraca locomotion investigations. Measurement: Journal of the International Measurement Confederation, 110, 249-256. doi:10.1016/j.measurement.2017.07.004
Author Response
Response to Reviewer 3 comments
Dear Colleagues,
The idea on which the paper is based is simple and not new as multi-parameter environmental control units are widespread in many cities. In fact, many of the political choices that affect citizens' lives depend on their measurements, such as the closure to traffic of cities or parts of the city with limited traffic areas or the adoption of alternating license plates, etc. Nevertheless, the technical challenge faced by this work from an electronic point of view with the development of a new economic control unit, from a more measuring point of view, with the management of the calibration of the system and from a less prosaic point of view of the collection and management of data coming from many control units distributed in the city for a long time, must certainly be rewarded. I add that the article is very clear, never boring, very well balanced in its subsections. The bibliography is valid and it is in particular for the measurement aspects related to the calibration of these control units which are by no means trivial. Even the technical choices relating to the sensor nodes fall within the state of the art of these devices.
Point 1: As far as I'm concerned, the article can be accepted in this way, from a purely academic point of view, I allow myself to point out three publications that could be of interest to the Authors and to be reasonably considered for the bibliography in the introduction:
Pasquali, V., D'Alessandro, G., Gualtieri, R., & Leccese, F. (2017). A new data logger based on raspberry-pi for arctic notostraca locomotion investigations. Measurement: Journal of the International Measurement Confederation, 110, 249-256. doi:10.1016/j.measurement.2017.07.004
Response 1: The referee mentioned three publications, but there is only one reference listed. We consider that the listed citation addresses the design of data logging for animal behavior, which is way out of the scope of this paper, so we decided not to include this citation in the revised manuscript.

Reviewer 4 Report
As it appears from studying the paper entitled “Calibration of sensor network for outdoor measurement of PM2.5 on high wood-heating smoke in Temuco City”, its content is quite interesting and addresses to the readership of the Journal.
The state-of-the-art cited in the introduction section is quite adequately described and critically analyzed.
The methodology which is followed in this work is adequately defined and assessed, permitting other researchers to reproduce certain aspects of the results. Additionally, the methodology analysis, as well as the assessment of the results are enriched with an efficient number of properly presented figures, tables and charts.
Nevertheless, in the paper there are some issues that should be attended:
1. The authors should provide more feedback on the research problem so as to fully succeed in justifying the novelty and scientific contribution of their work in the introduction section. Also, adding also a paragraph at the end of the introduction section that will refer to the structure of the article would be for the benefit of the readership.
2. Although the paper includes a Discussion and Conclusions section, its content is rather brief. It is suggested the results to be thoroughly interpreted in perspective of the working hypotheses, and the findings of the research. In addition, their implications should be discussed in the broadest context possible. Finally, the outcomes of the research should be concluded and some details about the future directions it would be interesting to be pointed out.
Finally, the paper is well-structured in general and written in appropriate English language according to the standards of the Journal, however some spell-checking is needed.
The paper is written in appropriate English language according to the standards of the Journal, however some spell-checking is needed.
Author Response
Response to Reviewer 4 comments
Comments and Suggestions for Authors
As it appears from studying the paper entitled “Calibration of sensor network for outdoor measurement of PM2.5 on high wood-heating smoke in Temuco City”, its content is quite interesting and addresses to the readership of the Journal.
The state-of-the-art cited in the introduction section is quite adequately described and critically analyzed.
The methodology which is followed in this work is adequately defined and assessed, permitting other researchers to reproduce certain aspects of the results. Additionally, the methodology analysis, as well as the assessment of the results are enriched with an efficient number of properly presented figures, tables and charts.
Nevertheless, in the paper there are some issues that should be attended:
Point 1: The authors should provide more feedback on the research problem so as to fully succeed in justifying the novelty and scientific contribution of their work in the introduction section.
Response 1. The referee suggests that we should provide more feedback on the research problem so as to fully succeed in justifying the novelty and scientific contribution of their work in the introduction section. We have modified the text in the introduction (line 92).
“The novelty of this paper and their main contribution is that we deployed a calibrated LCPM sensor network over a city with high contamination episodes, and this network allows us to analyze the behavior of the pollutants in an extensive part of the city over the entire event of high contamination episode. This network will allow us to measure the pollutants in the city, evidencing the differences in particle matter concentration within high contamination episodes simultaneously in different sectors.”
Point 2: Also, adding also a paragraph at the end of the introduction section that will refer to the structure of the article would be for the benefit of the readership.
Response 2: The following text showing the document’s structure has been included at the end of the introduction section (line 97).
“The paper is structured as follows: Firstly, Section 1 furnishes an overview of the current research status on obtaining information on intra-urban space-time variation, which allows extracting sectorized information on PM2.5 pollution and depicts the objective of deploying and calibrating a low-cost sensor network platform in the city of Temuco, Chile. Then, in Section 2, we show the methods, and Section 3, focuses on depicting a novel calibration method for low-cost particle matter sensors based on a nonlinear calibration model. Subsequently, in Section 4, we address the case study, presenting the deployment of the LCPM sensor network in Temuco City. As the main result, we show the spatio-temporal analysis of PM2.5 in three consecutive days, in which we can observe the accumulation of pollution and the washout due to the rain. Lastly, in Section 5, results and conclusions show the experience deploying this sensor network and the calibration method, including limitations and future research directions.”
Point 3: Although the paper includes a Discussion and Conclusions section, its content is rather brief. It is suggested the results to be thoroughly interpreted in perspective of the working hypotheses, and the findings of the research. In addition, their implications should be discussed in the broadest context possible.
Response 3: In the results and conclusion section, we had this paragraph:
“The main contribution of this paper is to review the dynamic behavior of the pollutants in the whole city over the entire event of high contamination episode”.
which was complemented by adding the following paragraph (line 275):
“This spatio-temporal analysis provides an insight into how the contaminants move in Temuco city along time on episodes of high contamination and cleansing due to rain.”
Point 4: Finally, the outcomes of the research should be concluded and some details about the future directions it would be interesting to be pointed out.
Response 4: At the end of the results and conclusion, we have added the following paragraph to provide guidelines for future work:
“The study showed that relevant data to understand the movement of spatio-temporal behavior of the contaminants could be acquired using calibrated LCPM sensor networks. The future challenge is to keep running this LCPM sensor network, increasing the number of nodes and replicating this setup in other cities with their features, such as population density, network coverage, and topography. It is relevant to study strategies to keep involving the city's inhabitants in the issue of deploying the sensor network with physical spaces for LCPM sensor installation and using their home's connectivity internet.”
Point 5: Finally, the paper is well-structured in general and written in appropriate English language according to the standards of the Journal, however some spell-checking is needed.
Comments on the Quality of English Language
The paper is written in appropriate English language according to the standards of the Journal, however some spell-checking is needed.
Response 5: We have revised the spell-checking according to the reviewer’s suggestion.
